# Study of Surface Modifications of Textile Card Clothing (AISI 1065 Alloy) by Laser Shock Peening

**DOI:** 10.3390/ma16113944

**Published:** 2023-05-25

**Authors:** Praveena Dhakshinamoorthy, Krishnan Harihara Subramanian, Karthik Kannan, Geetha Palani

**Affiliations:** 1Department of Physics, SRM Valliammai Engineering College, Kattankulathur, Kanchipuram 603203, India; praveenasnathan@gmail.com; 2Chemical Sciences Department and The Radical Research Center, Ariel University, Ariel 40700, Israel; karthikkannanphotochem@gmail.com; 3Institute of Agricultural Engineering, Saveetha Institute of Medical and Technical Sciences, Chennai 602105, India

**Keywords:** metallic card clothing, ablative layer, AISI 1065, bainite structure, laser shock peening, carbides

## Abstract

AISI 1065 is a carbon steels that is widely used in manufacturing industrial components owing to its high tensile strength and wear resistance. One of the major applications of such high-carbon steels is the manufacturing of multipoint cutting tools for materials such as metallic card clothing. The quality of the yarn is determined by the transfer efficiency of the doffer wire, which depends on its saw tooth geometry. The life and efficiency of the doffer wire depends on its hardness, sharpness, and wear resistance. This study focuses on the output of laser shock peening on the surface of the cutting edge of samples without an ablative layer. The obtained microstructure is bainite, which is composed of finely dispersed carbides in the ferrite matrix. The ablative layer induces 11.2 MPa more surface compressive residual stress. The sacrificial layer acts as a thermal protectant by decreasing surface roughness to 30.5%. The sample with a protective layer has a value of 216 HV, which is 11.2% greater than that of the unpeened sample.

## 1. Introduction

AISI 1065 is a high-carbon steel noted for its high tensile strength and wear resistance. It finds applications in manufacturing components working in high-friction environments at high loads. The components are clamps, couplings, cutting tools, surgical knife blades, springs, washers, and gears [1]. One of the major applications of AISI 1065 high-carbon steel is in the manufacturing of multipoint cutting tools for materials like metallic card clothing generally used in the textile industry. Textiles are one of the major contributors to a country’s economy and involve the conversion of fiber to yarn and yarn to fabric [2]. There are many processes involved, among them carding is the main mechanical process that decides the quality of the yarn [3]. In carding, unorganized clumps of fibers are entangled into individual fibers by metallic card clothing, which is fronted by rows of teeth or small wire hooks. The blending of different fibers and colors is carried out during the carding process [4]. Carding is a mechanical process whereby a large cylinder made up of metal is rotated at different speeds. The surface of the cylinder is closely wound with a metallic wire with a saw tooth profile. This is called metallic card clothing or doffer wire [5]. The quality of the yarn is determined by the transfer efficiency of the doffer wire, which depends on its saw tooth geometry. The life and efficiency of the doffer wire depend on its hardness, sharpness, and wear resistance [6]. The doffer is a collector that should have a sharp tooth. The tip of the doffer wire must be harder than the rib. There should be a gradual decrease in hardness from the tooth’s tip to the rib for an effective carding process. A uniform hardness of the doffer wire leads to mounting problems. To increase the holding efficiency of the doffer, the surface of the tip of the doffer is made uneven by serrated tooth edges [7]. Small increases in doffer efficiency improve the quality and increase the carding production. The wear-induced loss of sharpness in the upper sections of the teeth during carding is a persistent technical problem [8]. 

When the doffer wire tooth is exposed to fibers for a long time, the edges become rounded due to wear, and the metallic card clothing eventually loses its capacity to grab fibers as a result of the high working angle on the tips of the wire teeth, causing a significant impact on the carding quality. The tip of the doffer wire is hardened through heat treatment and wear-resistant coatings after machining into a saw tooth profile to increase the surface hardness of the teeth. There are very few reports on surface modification of the edges of saw teeth. The methods widely used in hardening are flame hardening and induction hardening. J. Z. Lu et al. subjected doffer wire to a plasma surface strengthening process (double glow technology) and reported an increase in the hardness due to chromizing coating, which improved the wear resistance [9]. Zhang et al. reported that the application of laser shock peening on a circular saw tooth blade improved the natural frequency, which increased the critical speed and enhanced the working condition [10]. 

Hardening the saw tooth by employing heat treatment not only improves the hardness of the material but also reduces its toughness. Heat treatment makes the material soft, and the tool wears excessively. With increased toughness, the saw tooth becomes brittle. To overcome this difficulty, Salimianrizi, E. et al. suggested a method of increasing the hardness of the saw tooth by induction hardening. In induction hardening, a high-frequency induction current rapidly heats the tooth passing through the material [11]. Hoppius et al. found that pulsed laser radiation increased the microhardness of a saw tooth made of tool steel 9CrV [12]. Laser shock peening (LSP) is more prominent than conventional methods, as it can be concentrated in a particular area without causing any distortion to the unexposed area. Changyu Wang and co-workers carried out trials of LSP by varying the energy densities on AISI 420 martensitic stainless steel and reported that LSP effectuated corrosion fatigue [13]. Laser shock imprinting (LSI) morphology was studied on 316 stainless steel surfaces, and subsequent LSI impacts were found to reduce surface deformation [14]. Luca Petan et al. examined the impact of LSP on the surface integrity traits of 18% Ni margining steel. The ablative properties of laser pulses caused an increase in the surface hardness. The enhancement of surface hardness and residual stress was due to plastic deformation caused by the LSP process [15]. The improvement of stainless steel by applying the LSP process was investigated by Spirit and coworkers. The specimen’s fatigue life was prolonged by the compressive residual stress imparted by shock waves [16]. Figure 1 shows a schematic diagram of the carding procedure.

The continuous loads on the doffer gradually weaken the structure of metallic card clothing, which, in turn, affects the efficiency of the doffer. Therefore, constant replacement of the metallic card clothing is required, which increases the production cost and downtime. This was the main motivation for this paper. The longevity and service life of the cutting edge can be increased by increasing the hardness of the cutting edge. The key goal of this paper is to study the changes in microstructure, hardness, and compressive residual stresses after irradiating the cutting edge with laser shock pulses.

## 2. Materials and Methods

### 2.1. Materials

In this study, metallic card clothing made of AISI 1065 high-carbon steel was subjected to LSP. The chemical composition of the metallic card clothing was analyzed by a chemical OES method (OES-Foundry Master, WAS, Rosengarten, Germany) and is listed in Table 1, with the mechanical properties of AISI 1065 high-carbon steel tabulated in Table 2. The samples were cleaned with deionized water and acetone, and the surface was polished with emery papers of different grits to achieve a mirror effect in order to ensure that the surface exposed to the laser didn’t have any irregularities [17]. The wire electrode cutting method was employed to cut the samples to 3.5 cm each.

### 2.2. Methods

A Nd:YAG laser (SPECTRO SL905 TQ, AL Dalton Ltd., Nottingham, UK) with a 1064 nm wavelength beam, a 10 Hz pulse frequency, 0.8 mm beam diameter on the sample, and 10 ns pulse duration was utilized for this experiment. The experimental setup of the laser peening process and plasma formation is shown in Figure 2a,b.

The surfaces of all samples were polished before testing and cleaned with acetone. The surface of the sample was coated with commercially available black paint that served as an absorbing layer and was immersed in water at a depth of 2 to 3 mm, which served as a confining layer. In order to prevent metallic materials from vaporizing and evaporating, black paint and aluminum foil are frequently used as absorbing layers. Water or BK7 optical glass is typically used as the confining layer to restrain the expansion of the laser-induced plasma and, as a result, increase the shockwave pressure and duration [18]. In LSP, a sacrificial layer (black paint) is irradiated with a high-energy-density, short-duration pulsed laser through a constraint layer (water). The sacrificial layer absorbs the laser energy and forms rapidly expanding plasma, which causes a strong shockwave to propagate into the work piece. When the shockwave’s power exceeds the material’s dynamic yield strength, plastic deformation results. The process also induces high-density dislocation and compressive residual stress to increase the material’s strength [19].

## 3. Results and Discussions

### 3.1. Microstructural Analysis

Laser shock peening brings about changes in the surface and subsurface area of the sample. The surfaces of the samples were prepared and examined by a metallurgical microscope. The doffer wires, as in the received condition, have fine grains of tempered martensite, which exhibit a body-centered tetragonal (BCT) crystal system. The martensite phase is brittle; slight quenching of the martensite softens the material, reducing its brittleness. The hardness of the material also decreases. Tempered martensite more or less the same hardness as martensite with improved ductility and toughness. When the samples are subjected to laser shock peening the obtained microstructure is bainite, which is finely dispersed carbide in the matrix of ferrite. The formation of bainite is due to high-density dislocations as a result of plastic relaxation effects caused by shock waves. The presence of bainite increases impact toughness and ductility more than tempered martensite. The strength and hardness of a material are controlled by refinement of subunits of bainite microstructure, which increases the hardness of the material without compromising the ductility. The ultimate tensile strength and yield strength of the sample are both increased by the presence of fine carbides. Fine carbides obstruct the path of dislocations and enhance the work hardening. As the carbides are finer, the fracture strength of the material increases due to an increase in dislocation density [20,21,22,23,24,25]. Figure 3 shows the microstructure of unpeened and laser-shock-peened samples with and without black paint.

### 3.2. Residual Stresses

The plastic deformation brought about by shock waves is what leads to the residual compressive stresses in a metal. Elastically stretching plastic deformation is caused by extremely high strain rates. The compressive residual stresses are balanced by the tensile stresses below the surface of compressive stresses [26]. The residual stress arising from the surface treatment process such as laser peening retards the fatigue life and increases the hardness of the sample [27]. The X-ray diffraction sin^2^φ method (proto iXRD instrument, IIT, Madras, India) was used to measure the residual stresses of peened and unpeened samples. A 2.291 Å wavelength from a Cr K_α_ X-ray source with a round aperture with a 2 mm diameter was utilized for the residual stress measurements. Reflections from the (211) plane at a Bragg angle of 156.41° were used for the analysis [28]. The specimens were cut into 10 mm lengths by an electrical discharge machining (EDM) wire so that the entire tooth and the rib were covered. The absorptive layer was removed by cleaning the surface with acetone. Figure 4 shows the values of residual stress.

The initial residual stresses of the unpeened sample were +2.5 ± 8.5 MPa due to the effects of the manufacturing process and heat treatment. The irradiation of laser pulses on the sample with and without black paint created compressive residual stresses of −171.2 ± 8.0 MPa and −182.4 ± 7.2 MPa, respectively. The ablative layer induced 11.2 MPa more compressive residual stress on the surface, which is in agreement with the reports of Nikola. The ablative layer enhances the depth of residual stress [29]. The results show that the tensile stress in the material was converted into compressive residual stress. An imbalance was created between the permanent plastically deformed region and the elastically deformed regions beneath. This imbalance was the source of residual compressive stresses, which were confined to a thin layer. This compressive stress was balanced by the tensile residual stress extending beneath [30].

### 3.3. Surface Roughness 

The surface roughness of the laser-shock-peened and unpeened samples was measured (Mitech Surface Roughness Tester MR200, Unique measurement service, Bangalore, India). For each sample, three different positions on the surface were selected, and the arithmetic roughness (R_a_, μm) was measured (Table 3) (Figure 5). It was found that the surface roughness of the samples was affected, as the exposure of the sample to the laser beam ablated the surface and induced plastic deformation in the sample. The results confirm that the surface roughness of the peened sample increased considerably, which is in remarkable agreement with the findings of previous research [31]. The surface roughness plays a significant role in industrial components. The surface roughness of the unpeened specimen before LSP was 1.177 μm. The surface roughness of the sample coated with a sacrificial layer increased by 14% to 2.544 μm. The surface roughness value of the specimen without a sacrificial layer was found to increase by 44% from 2.544 to 3.461 μm. Comparison of the surface roughness for the specimens with and without coating showed that the sacrificial layer acted as a thermal protectant by decreasing the surface roughness to 30.5%.

### 3.4. Microhardness

The changes in the hardness of the sample were due to the shock waves that were produced when the pulsed laser caused the high-pressure plasma to expand. The shock waves produced by plasma create grain dislocations, which alter the microhardness of the sample, which is the after effect of plastic deformation [32]. Hardness studies were carried out on the samples. Microhardness is a property indicating the ability of a material to resist indentation; the higher the hardness value, the greater the resistance to wear [33]. LSP is a process that produces a high strain rate and alters the microhardness of samples by inducing the dislocation. Density was determined using a Micro Vickers hardness tester with a load of 100 g and holding time of 10 s. The hardness was examined for every 0.2 mm from the surface as a function of depth. It was observed that the hardness of the peened samples increased in comparison with that of unpeened samples. The obtained results are consistent with the results reported by Arpit Siddaiah et al. [34]. The hardness of single side-peened samples was found to decrease gradually with depth from the surface. Microhardness decreases as a result of water confinement absorption, which lowers the laser intensity and absorbs surface-level laser pulse energy [35]. The variation in the hardness is shown in Figure 6, which shows an increase in the hardness value due to LSP. The sample with a protective layer (black paint) had a value of 216 HV, which is 11.2% greater than that of the unpeened sample. When the samples coated with and without a sacrificial layer were compared, the hardness value increased by 7 HV, from 209 HV with a coating to 216 HV without a coating.

## 4. Conclusions

LSP techniques were used on high-carbon AISI 1065 steel both with and without an ablative coating. The effects of LSP treatment on microstructure, microhardness, and residual stresses were examined. 

The microstructure of the laser-peened samples with and without sacrificial layers was analyzed and compared with that of the unpeened sample. LSP changes the microstructure of the base sample from martensite to a bainite phase that has finely dispersed carbides. The martensite phases increase the brittleness of the sample, which greatly affects the carding process. The bainite phase surface increases the hardness without sacrificing the toughness. The microstructure of both laser-peened samples with and without black paint has a bainite phase; this suggests that the tip of the doffer wire can be irradiated by laser pulses without black paint; doffer wire is used in large textile industries.The shock waves induced on the surface cause plastic deformations in the surface and subsurface areas that induce compressive residual stresses. These compressive residual stresses increase dislocation density, which blocks dislocation movement at the grain boundary, improving fatigue life by enhancing work hardening. An increase in residual stresses of about 194% occurred after LSP processes. The ablative layer increased the residual stresses by 6.3%.The Vickers hardness test shows that LSP increased the hardness of the sample by 11.2% when compared with the unpeened sample. The hardness value increased by 7 HV between the samples with and without a sacrificial layer, from 209 HV with a coating to 216 HV without a coating.The LSP process increased the surface roughness of the sample without coating by 44% and that of the sample with black paint by 14%, confirming the protective role played by the ablative layer in reducing laser ablation. LSP with a protective layer is a mechanical process, whereas without an ablative layer, the surface absorbs the thermal energy of the laser, so it is a thermo mechanical process. The increase in the surface roughness of the cutting edge adds an advantage to metallic card clothing; it increases the holding capacity of the fiber, which plays the same role as horizontal serrated doffers.

## Figures and Tables

**Figure 1 materials-16-03944-f001:**
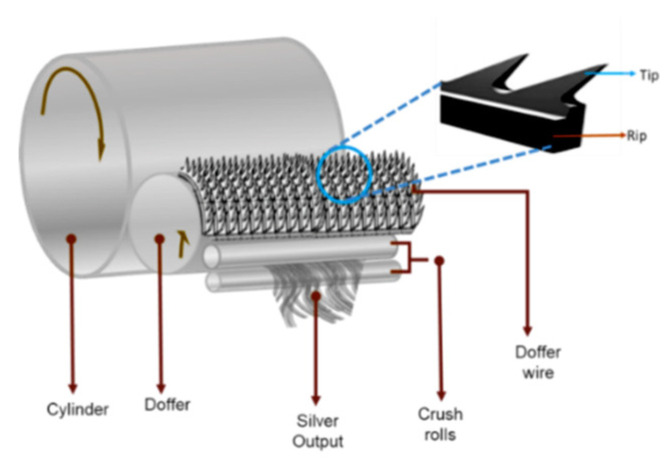
Schematic diagram of carding and doffer wire.

**Figure 2 materials-16-03944-f002:**
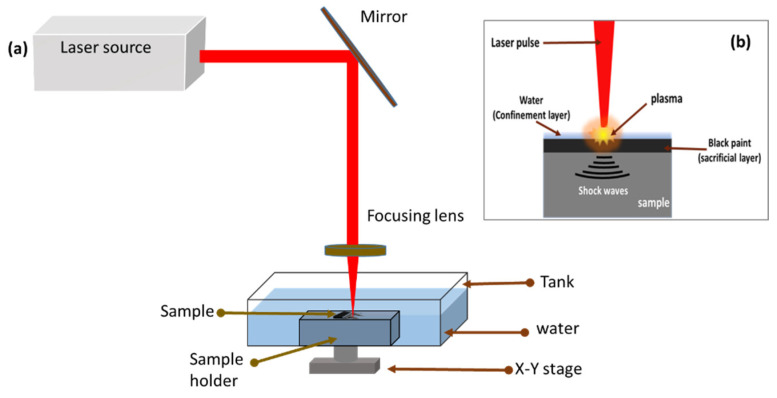
(**a**) Experimental setup for laser shock peening; (**b**) plasma formation on the surface.

**Figure 3 materials-16-03944-f003:**
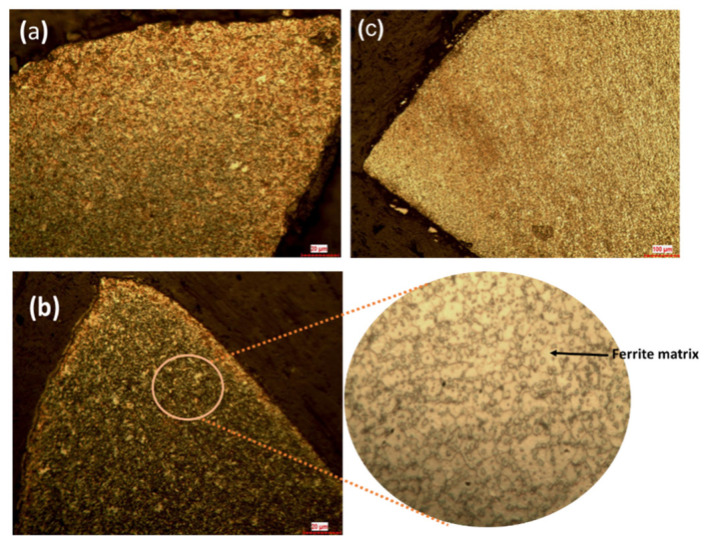
Microstructure of samples subjected to LSP: (**a**) raw material; (**b**) sample with paint (in site–ferrite matrix); (**c**) sample without black paint.

**Figure 4 materials-16-03944-f004:**
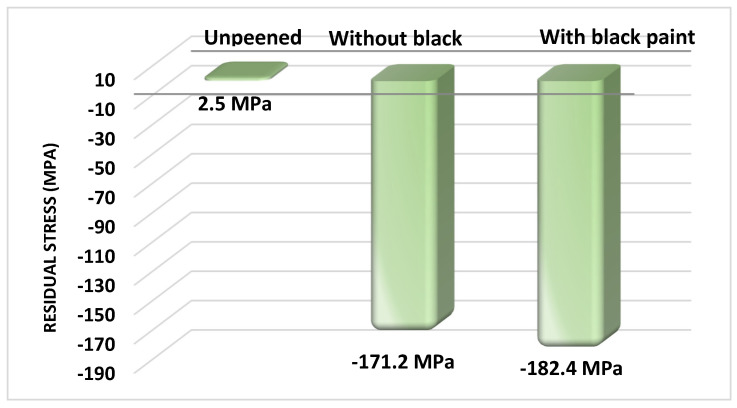
Residual stress of unpeened and laser-shock-peened samples.

**Figure 5 materials-16-03944-f005:**
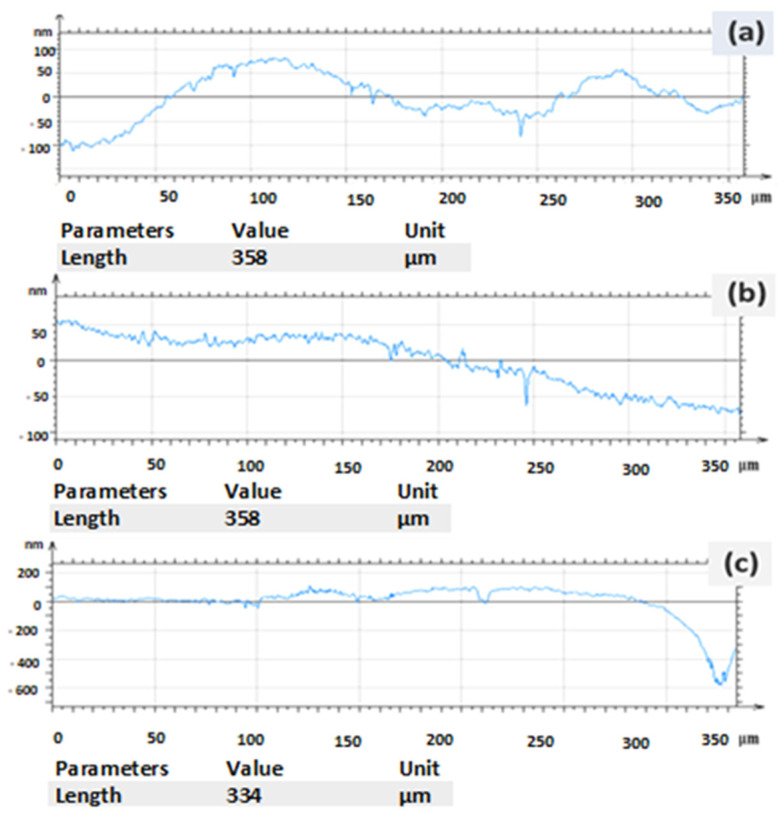
Surface roughness of (**a**) unpeened and (**b**) laser-peened samples with black paint and the (**c**) laser-peened sample without black paint.

**Figure 6 materials-16-03944-f006:**
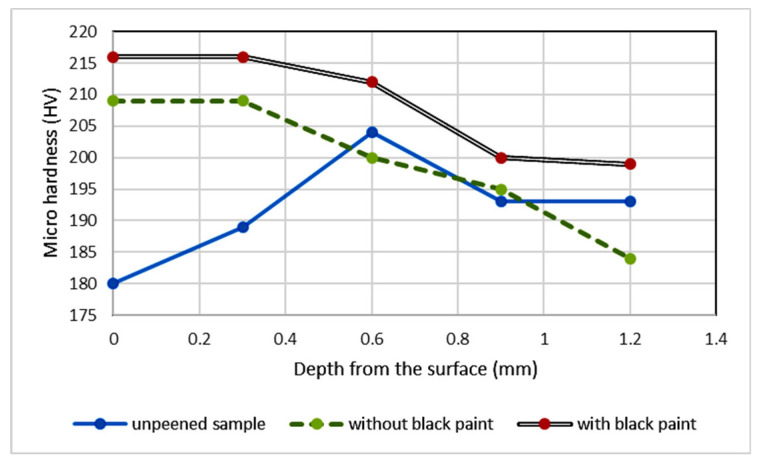
Microhardness of unpeened and laser-shock-peened samples.

**Table 1 materials-16-03944-t001:** Chemical composition of AISI 1065 high-carbon steel (wt.%).

Composition	C	Mn	Si	S	P
Percentage (wt.%)	0.64	0.77	0.205	0.90	0.035

**Table 2 materials-16-03944-t002:** Mechanical properties of AISI 1065 high-carbon steel (wt.%).

Density (g/m^3^)	Modulus of Elasticity (GPa)	Poisson’s Ratio	Ultimate Tensile Strength (MPa)	Tensile Yield Strength (MPa)	Elongation (%)
7.85	200	0.27	635	490	18

**Table 3 materials-16-03944-t003:** Surface roughness (R_a_) values (μm).

Si. No	Unpeened	Laser-Shock-Peened with Paint	Laser-Shock-Peened without Paint
1	1.241	2.487	3.025
2	1.033	2.612	3.841
3	1.257	2.533	3.516
Average	1.177	2.544	3.46

## Data Availability

Data sharing is not applicable to this article, as no datasets were generated or analyzed during the current study.

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
