# Peer review of "Study of Surface Modifications of Textile Card Clothing (AISI 1065 Alloy) by Laser Shock Peening"

_materials, 2023, doi:10.3390/ma16113944_

Round 1
Reviewer 1 Report
The application of LSP on 1065 carbon steel for pontial application for the parts of metallic card clothing is rare and valuable work. The finding of microstructural modification, hardness and roughness with and without the ablative layer are helpful for the application of LSP to the industry.
The manuscript can be much more improved if the following revision is made:
- In the figure caption, please add anaysis method.
The main question addressed by the research are:
- Microstructural evolution and resulting mechanical property modification.
- How was the dislocation density increase detected?
The application of LSP on the parts of card clothing and resulting increasing surface roughness increasing the performance is of interest, unique and industrially applicable.
The manuscript can be drastically improved by adding microstructural analysis using such equipment as AFM, TEM, etc.
This manuscript adds to the subject area the following, when compared to already published materials:
- Effect of surface roughness and effect of that on the performance.
- Effect of the protective layer.
The authors should consider the effect of the severity of the peening, such as length of the treatment, in their paper.
The conclusion #1-3 seems to be week, while #4 is significant.
The references are appropriate.
- Figure 3: Length markers are not apparent.
AFM surface roughness will be a good addition.
- In Fig. 3, a1, b1, c1 and a2, b2, c2 are different in magnification without obvious region.
- Fig. 5-6: Readability can be improved by increasing the font size of the axis scales and make to overall figure more concise
- Add error bar in Fig. 6.
- English is good.
- There appears to be minors errors:
line 133: Microstructure--> microstructure
line 225: ashorizontal --> as horizontal
Author Response
Dear Reviewer,
Coauthors and I very much appreciated the encouraging, critical, and constructive comments given by the reviewers on this manuscript. We strongly believe that the comments and suggestions have increased the scientific value of the revised manuscript by many folds. We have taken them fully into account in revision. We are submitting the corrected manuscript with the suggestion incorporated into it.

Reviewer 2 Report
Dear authors, it was interesting to read your article because laser treatment of surface always has an interesting nuances influencing the structure. But I have few comments:
1. chapter 2.2. Methods needs to be improved with the more detailed explanation of laser peening process. In figure 2 we can see that samples was under the water during treatment, isn't it? Not clear what exactly palse of sample was treated. And Fig. 2 needs to be improved. May be it can be shown only actual area of treatment highlighting the important points.
2. In Fig.3 you mentioned "black paint microstructure". But, any explanation about the black paint meaning was presented before. You Need to discover what you mean under the "black paint" . An may be it will be good to put the references to fig 3 when describing the microstructure in chapter 3.1. It needs more clearly show the difference of A1 A2, B1 B2 and D1 D2.
As I understood from table 3 black paint means coating for higher absorption of laser irradiation, isn't it? If so, you need to open that in methodology, because now I understood, that black paint is the elements in microstructure of steel.
3. You describing the residual stress in surface after LP. (chapter 3.2. and fig. 4) But it not clear where such stress was measured? Is that stress (that values) in top layers or distributed in volume?
4. Citation: "The comparison of surface roughness for the speci- 174 men with and without coating shows that the sacrificial layer acts as a thermal protectant 175 by decreasing surface roughness to 30.5%" What you mean saying "Coating"? Is it extra coating, or it is strengthened layer? If it extra coating it needs to be mentioned in chapter of materials and methods.
5. You need to improve the title of Fig.5 with indicatingthe what is a, b and c
6. Fig 6. It will be better to choose different mark shape, not only the color of lines. O take out the gray background.
7. Conclusions. Here you talk about the ablative coating. So the methodology chapter needs to be improved.
1. Fig.2
Author Response
Dear Reviewer,
My co-authors and I very much appreciated the encouraging, critical, and constructive comments given by the reviewers on this manuscript. We strongly believe that the comments and suggestions have increased the scientific value of the revised manuscript by many folds. We have taken them fully into account in the revision. We are submitting the corrected manuscript with the suggestion incorporated into it.
